# The Longevity of Fruit Trees in Basilicata (Southern Italy): Implications for Agricultural Biodiversity Conservation

Jordan Palli [1,2], Michele Baliva [1,2], Franco Biondi [3], Lucio Calcagnile [4], Domenico Cerbino [5], Marisa D'Elia [4], Rosario Muleo [2], Aldo Schettino [6], Gianluca Quarta [4], Nicola Sassone [5], Francesco Solano [1,2], Pietro Zienna [5] and Gianluca Piovesan [1,*]

1  Department of Ecological and Biological Sciences (DEB), University of Tuscia, 01100 Viterbo, Italy
2  Department of Agriculture and Forest Sciences (DAFNE), University of Tuscia, 01100 Viterbo, Italy
3  DendroLab, Department of Natural Resources & Environmental Science, University of Nevada, Reno, NV 89557, USA
4  Centre of Applied Physics, Dating and Diagnostics (CEDAD), Department of Mathematics and Physics "Ennio De Giorgi", University of Salento, 73100 Lecce, Italy
5  Lucan Agency for the Development and Innovation in Agriculture (ALSIA), via Annunziatella 64, 75100 Matera, Italy
6  Ente Parco Nazionale del Pollino, Complesso Monumentale Santa Maria della Consolazione, 85048 Rotonda, Italy
*  Correspondence: piovesan@unitus.it

**Abstract:** In the Mediterranean basin, agriculture and other forms of human land use have shaped the environment since ancient times. Intensive and extensive agricultural systems managed with a few cultured plant populations of improved varieties are a widespread reality in many Mediterranean countries. Despite this, historical cultural landscapes still exist in interior and less intensively managed rural areas. There, ancient fruit tree varieties have survived modern cultivation systems, preserving a unique genetic heritage. In this study, we mapped and characterized 106 living fruit trees of ancient varieties in the Basilicata region of southern Italy. Tree ages were determined through tree ring measurements and radiocarbon analyses. We uncovered some of the oldest scientifically dated fruit trees in the world. The oldest fruit species were olive (max age 680 $\pm$ 57 years), mulberry (647 $\pm$ 66 years), chestnut (636 $\pm$ 66 years), and pear (467 $\pm$ 89 years). These patriarchs hold a unique genetic resource; their preservation and genetic maintenance through agamic propagation are now promoted by the Lucan Agency for the Development and Innovation in Agriculture (ALSIA). Each tree also represents a hub for biodiversity conservation in agrarian ecosystems: their large architecture and time persistence guarantee ecological niches and micro-habitats suitable for flora and fauna species of conservation significance.

**Keywords:** fruit tree age; olive; chestnut; pear; mulberry; natural heritage; cultural landscape; dendrochronology; radiocarbon; agriculture biodiversity

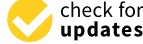



## 1. Introduction

In the Mediterranean basin, agriculture, livestock herding, timber harvesting, and other forms of land use have shaped the rural environment and generated a mosaic of semi-natural cultural landscapes. For millennia, natural resource utilization has impacted wild ecosystem distribution and function at different spatial scales [1]. Such long-term reclamation of space for activities related to human sustenance has inevitably led to natural habitat fragmentation and biodiversity loss [2]. In modern times, industrialization and technological developments have modified agricultural systems to the point that currently, productivity targets are beyond the limits of environmental sustainability [3]. Intensive agriculture has also simplified the population structure of cultured plants, and agricultural biodiversity is being lost as well.

Countries characterized by an ancient land use history display a variety of different environments, spanning from intensively exploited agricultural areas to highly natural ecosystems [4]. The environmental heterogeneity that characterizes regions such as southern Italy was generated by historical phases of intense land exploitation followed by abandonment and rewilding [5]. Remote and formerly inaccessible mountain areas now feature natural looking old-growth ecosystems as a result of rewilding processes that started in late Medieval times [6]. Inner hills and plains in rural and marginal areas are still objects of traditional land use practices that maintain large landscape patches to a semi-natural managed status. These are called "historical landscapes" and host agricultural elements that survived modern cultivation systems.

One of the agricultural elements capable of crossing the centuries is fruit trees. Monumental fruit trees are common in Mediterranean historical landscapes and orchards [7–9]. Olive and chestnut megaflora examples have often been studied using anecdotal data or general size-age relationship equations to estimate their age [10,11]. Olive and chestnut were selected, cultivated, and spread by humans for centuries in the Mediterranean Basin [12,13]. Ancient breeds are an important part of the rural agriculture economy [14], and very old individuals are maintained because of their economic value. Besides olive and chestnut, Mediterranean historical landscapes host ancient varieties of many other fruits, such as pear, apple, mulberry, cherry, service, and walnut [7]. These fruit trees are less frequently found in monumental sizes, so they draw little attention in the study of longevity. Famous grooves or ancient orchards are the exceptions [15,16].

Ancient trees underlie numerous ecosystem services [17] and preserve a unique genetic diversity [18,19], to be acknowledged and defended as an intergenerational heritage. They bear witness to the local cultural and landscape history and are a hub for biodiversity conservation in agrarian ecosystems. Old fruit trees are present from seminatural to urban ecosystems, and their survival depends on human activities as well as their own resilience against threats and disturbances. The abandonment of agricultural marginal lands, coupled with agricultural intensification and urbanization/urban sprawl, is endangering the survival of old-fruit trees and their transfer to future generations. Therefore, it is urgent to plan land use for the conservation of such old trees and for restoring connectivity in economic land uses in the eco-cultural context of rural areas.

Here, we present the distribution and attributes of 106 living old-fruit trees in southern Italy (Basilicata), some of which have survived over multiple centuries. The age of monumental fruit trees is often mystified and/or assumed on non-scientific bases, such as historical narratives, portraits, or other anecdotal data. Determining the age of monumental fruit trees with scientific methods is challenging because of rotten wood, growth anomalies and false rings. In this study, we aim at (i) determining stem ages of mature and old fruit trees with integrated tree ring and radiocarbon methods that overcome difficulties related to monumental sizes and hollowed tree stems; (ii) assessing growth patterns of old fruit trees with respect to younger ones; (iii) describing the environment in which old fruit trees grow; and (iv) discussing implications for conservation biology and sustainable management of agricultural lands. We also argue why and how to conserve their genotypes through specific initiatives that involve local communities, and we present the ongoing efforts promoted by the Lucan Agency for the Development and Innovation in Agriculture (ALSIA) to disseminate the value of ancient varieties of fruit trees for a sustainable future.

## 2. Materials and Methods

### 2.1. The Survey

The survey of fruit trees was carried out in southern Italy, in the region of Basilicata, (Figure 1) by trained ALSIA technicians. Old trees are scattered across marginal agricultural fields or orchards, often within private properties. The collaboration between owners, keepers and farmers was of crucial importance for a successful survey. ALSIA's specialized operators were tasked with the inspection, identification, description, mapping and sampling of putative old and ancient fruit trees reported by local people. Potential

old trees were recognized based on their structural features, with special attention paid to crown and stem characteristics (e.g., large diameter, gnarled/twisted shape). Each tree was measured, georeferenced, assigned to a species, and to a variety when possible. Measures included DBH, height, canopy size and shape, as well as the presence of dead branches and the overall health condition of the tree. Environmental features were also recorded, such as the environmental context (e.g., urban, peri-urban, agricultural), elevation, slope, topographic exposure, soil rockiness, soil depth and profile.

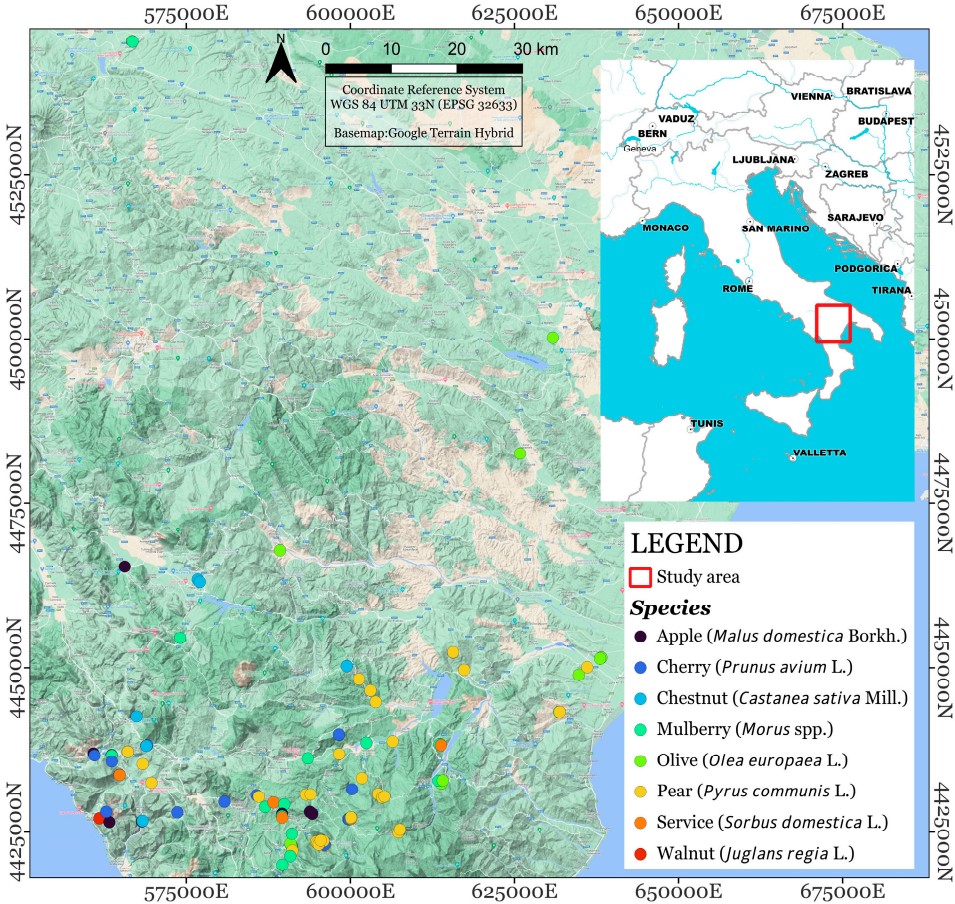

**Figure 1.** Distribution map of fruit tree sites (color-coded) included in this study. Basemap credit: Google Terrain Hybrid 2023; national boundaries are provided by Natural Earth (https://www. naturalearthdata.com/, accessed on 20 January 2023).

### 2.2. Longevity, Growth and GIS Analysis

Tree ages were determined through tree ring measurements and radiocarbon analysis. The method was chosen based on the type of sample and the tree species. Tree ring methods were preferred for deciduous species and when the tree stem was healthy and a complete wood increment core could be obtained. Radiocarbon analyses were used when stems were hollow or rotten. Radiocarbon dating can provide accurate tree ages with reduced confidence intervals by wiggle matching between two adjacent tree rings [20–22]. Wood cores were extracted from intact and healthy stems using a Pressler-type borer at or near 1.3 m above the ground. Trees with a hollow or rotten trunk were sampled by taking wood fragments with a scalpel from their innermost and basal part, i.e., the closest portion to the stem pith and/or root collar. Presumed ancient and mature trees were both sampled to compare growth trends during two different stages of the ontogenetic cycle.

A total of 70 trees presented a healthy stem and were sampled for dendrochronology analyses. Cores for tree-ring measurements were surfaced, polished and measured to the nearest 0.001 cm through the CCTRMD (Computer Controlled Tree Ring Measurement

Device) [23] and further processed through the software CATRAS (Computer Aided Tree Ring Analysis) [24]. Cores that included the stem pith were used to investigate DBH-age relationships and to reconstruct the DBH growth history with the aim of assessing long-term growth rates and patterns in fruit trees.

An additional set of 33 monumental trees presented a hollow or rotten stem; they were sampled for radiocarbon analyses. One sample for each tree was sent to the laboratory. Radiocarbon analyses were performed through Accelerator Mass Spectrometry (AMS) at the Center of Applied Physics, Dating and Diagnostics (CEDAD) of the University of Salento, Italy [25]. Radiocarbon dates were calibrated in OxCal with the IntCal20 calibration curve [26] using the 2σ confidence interval.

Coordinates of fruit trees were used to perform GIS analysis and assess the topographic position of trees (elevation, slope, aspect) and the land use/land cover features of the area in which they grow. These analyses were then used to discuss longevity and growth patterns of fruit tree species.

The complete method workflow is displayed in the flow chart in Figure 2.

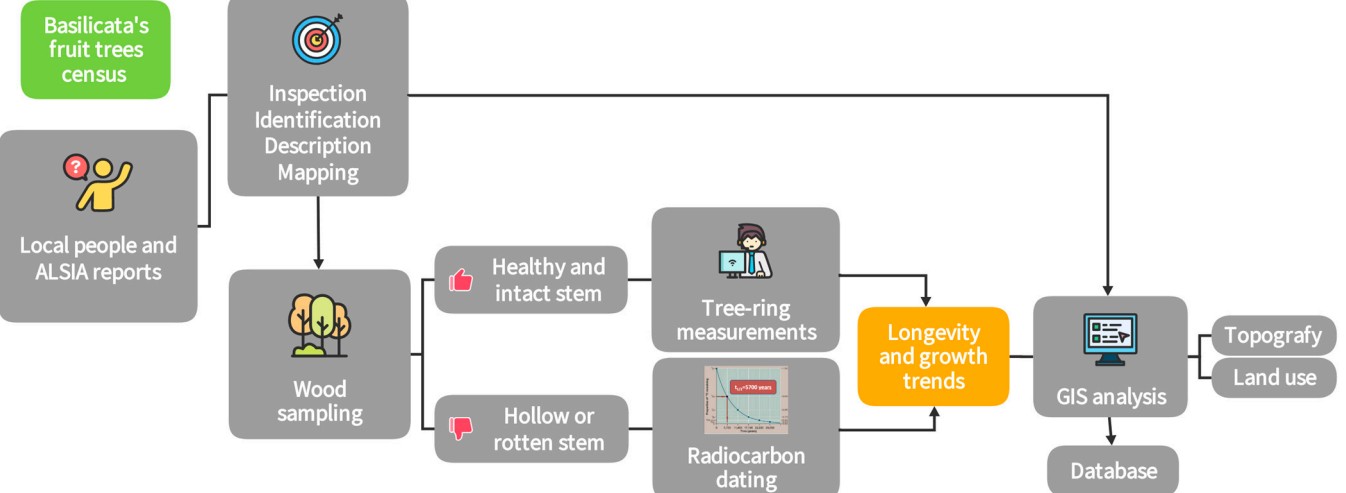

**Figure 2.** Flow chart describing the methods workflow. Reports from local people and ALSIA allowed for the inspection, identification, description, mapping, and wood sampling of old and mature fruit trees in Basilicata. Wood samples were tree-ring measured or radiocarbon dated based on stem morphology and wood condition. Longevity and growth trends assessed from tree ring and radiocarbon data were incorporated into the GIS environment to create an informative database and analyze topography and land use features. The flowchart was developed through the software GitMind, Ver. January 2023 (https://gitmind.com/, accessed on 10 February 2023).

## 3. Results

### 3.1. The Census of Fruit Patriarchs in Basilicata

The census of fruit tree patriarchs in Basilicata identified 106 individuals from 9 species, which have been georeferenced and mapped (Figure 1). The species involved in the study were: *Castanea sativa* Mill. (chestnut), *Juglans regia* L. (walnut), *Malus domestica* Borkh. (apple), *Morus alba* L. and *Morus nigra* L (grouped as mulberry), *Olea europaea* L. (olive), *Prunus avium* L. (cherry), *Pyrus communis* L. (pear), *Sorbus domestica* L. (service).

Fruit trees were distributed from the lowland to the mountain elevation belt, and from plain topography to steep slopes. The majority of them were concentrated in the warmest aspects, i.e., from SE to SW (Table 1). Pear trees were found over the largest elevation range, from 43 to 1097 m a.s.l., and on the steepest slopes (63%; Table 1). Additionally, pears were the only trees located on all aspects, including the coldest ones, i.e., N, NE and NW (Table 1). Other fruit species were absent on north-exposed land (Table 1). Cherry, pear and walnut were found above 700 m a.s.l. on average (Table 1). Olives were recorded at a

lower mean elevation, 362 m a.s.l., although some individuals were found up to 800 m a.s.l. (Table 1).

**Table 1.** Topography features of fruit tree locations summarized by tree species. Elevation is expressed in meters above sea level (m a.s.l.). Slope is expressed in percentages. Aspect is divided into 8 cardinal directions and expressed as % of trees per direction.

| Species | Elevation (m a.s.l.) | | | | Slope (%) | | | | Aspect (%) | | | | | | | |
|---|---|---|---|---|---|---|---|---|---|---|---|---|---|---|---|---|
| | Mean | SD | Min | Max | Mean | SD | Min | Max | N | NE | E | SE | S | SW | W | NW |
| Apple | 628 | 163 | 433 | 842 | 18 | 10 | 2 | 29 | - | 14 | - | 14 | 29 | 43 | - | - |
| Cherry | 753 | 157 | 488 | 932 | 14 | 10 | 1 | 36 | - | - | - | 8 | 25 | 8 | 42 | 17 |
| Chestnut | 485 | 281 | 203 | 874 | 12 | 8 | 2 | 25 | - | 9 | 9 | 9 | 36 | 36 | - | - |
| Mulberry | 579 | 180 | 260 | 857 | 23 | 8 | 1 | 34 | - | - | 40 | 20 | 20 | 13 | 7 | - |
| Olive | 354 | 256 | 73 | 807 | 15 | 10 | 3 | 32 | - | - | 36 | 21 | 7 | 7 | 14 | 14 |
| Pear | 698 | 261 | 43 | 1097 | 22 | 9 | 9 | 63 | 5 | 8 | 13 | 15 | 5 | 20 | 25 | 10 |
| Service | 590 | 124 | 479 | 762 | 25 | 4 | 20 | 30 | - | - | 25 | 25 | 25 | 25 | - | - |
| Walnut | 704 | 347 | 325 | 1006 | 26 | 21 | 9 | 49 | - | - | 33 | - | - | - | 33 | 33 |

Most old fruit trees were located in heterogeneous agricultural areas, which include complex cultivation patterns, forming a mosaic landscape where cultivated lands mix with semi-natural ecosystems (Table 2). Olive and mulberry trees could also be found in the discontinuous urban fabric category (Table 2). Old olive and chestnut trees were rarely located in olive groves and chestnut stands, and many chestnut trees lie in the shrub and/or herbaceous vegetation association (Table 2). Cherry, walnut and service trees were often recorded in forests or transitional woodland categories (Table 2).

**Table 2.** Land use and land cover features of fruit tree locations. Land use and land cover data were extracted from the Corine Land Cover (2018 v. 2020_20u1). Numbers indicate the CLC category: 112 = Discontinuous urban fabric; 211 = Non irrigated arable land; 222 = Fruit trees and berry plantation; 223 = Olive groves; 242/243 = Heterogeneous agricultural areas; 3111 = Broadleaved forest (Holm oak and/or Cork oak); 3112 = Broadleaved forest (Turkey oak and/or Downy oak and/or Hungarian oak and/or Pedunculate oak); 3114 = Chestnut forest; 3115 = Beech forest; 321 = Shrub and/or herbaceous vegetation associations; 324 = Transitional woodland shrub.

| Species | Land Use/Land Cover (%) | | | | | | | | | | |
|---|---|---|---|---|---|---|---|---|---|---|---|
| | 112 | 211 | 222 | 223 | 242/243 | 3111 | 3112 | 3114 | 3115 | 321 | 324 |
| Apple | - | - | - | - | 86 | - | - | - | - | 14 | - |
| Cherry | - | - | - | - | 50 | 8 | 33 | - | - | - | 8 |
| Chestnut | - | - | - | - | 27 | - | 9 | 9 | 9 | 45 | - |
| Mulberry | 20 | 7 | - | - | 67 | - | 7 | - | - | - | - |
| Olive | 14 | 7 | - | 7 | 71 | - | - | - | - | - | - |
| Pear | 3 | 20 | 3 | - | 63 | - | 8 | - | - | 3 | 3 |
| Service | - | - | - | - | 75 | - | - | - | - | - | 25 |
| Walnut | - | - | - | - | 67 | 33 | - | - | - | - | - |

The oldest ages were determined through radiocarbon analyses (Table 3). A total of 11 samples dated after 1950 CE and were thus discarded from further analysis (Table 3). The olive tree of Ferrandina (Figure 3a) reached 680 ± 57 years in 2022 CE. It is the oldest tree of our census, followed by a mulberry tree of 647 ± 66 years and by the chestnut tree of Lagonegro (Figure 3c), with 636 ± 66 years. The oldest pear tree, 467 ± 89 years old, is located in San Severino Lucano (Figure 3b).

**Table 3.** Statistics of fruit trees analyzed through radiocarbon analyses. Modern radiocarbon dates (asterisks) were not included in the results.

| Species | No. Trees (Total) | No. Trees (After 1950) | DBH (cm) | | | Radiocarbon Age (Years) | | | Elevation (m a.s.l.) | | |
|---|---|---|---|---|---|---|---|---|---|---|---|
| | | | Mean | Max | Min | Mean | Max | Min | Mean | Max | Min |
| Chestnut | 2 | 1 * | - | 247 | - | - | 636 | - | - | 865 | - |
| Mulberry | 8 | 4 * | 73 | 88 | 59 | 381 | 646 | 214 | 586 | 824 | 351 |
| Olive | 14 | 3 * | 159 | 255 | 81 | 382 | 680 | 212 | 288 | 776 | 73 |
| Pear | 13 | 6 * | 88 | 134 | 61 | 336 | 467 | 160 | 738 | 1078 | 270 |

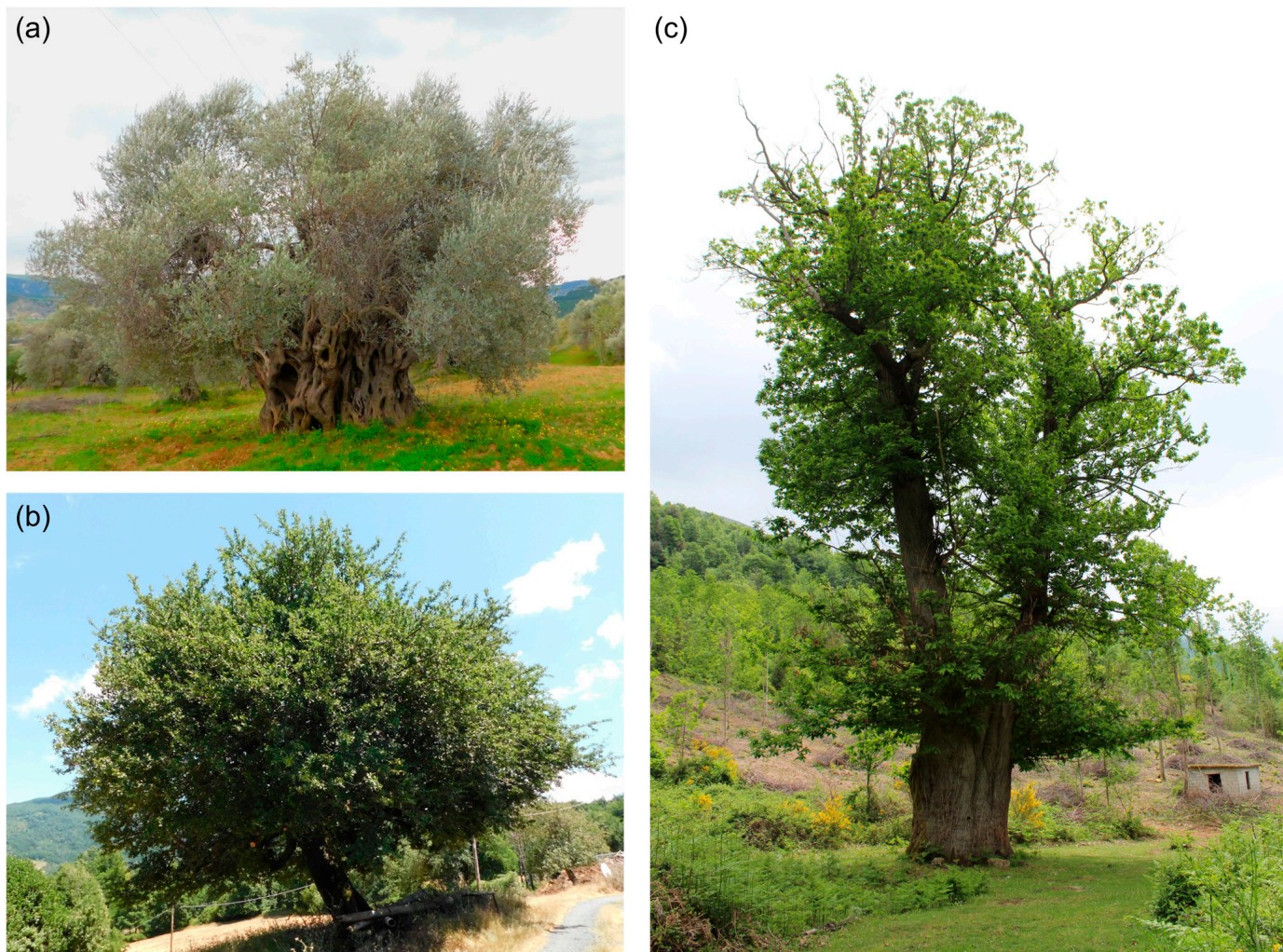

**Figure 3.** Very old individuals identified in this study. (**a**) The olive tree of Ferrandina: 680 ± 57 calibrated years in 2022. (**b**) The pear tree of San Severino Lucano: 467 ± 89 calibrated years in 2022. (**c**) The chestnut of Lagonegro: 636 ± 66 calibrated years in 2022. Photo credits: Domenico Cerbino.

The stem pith was reached in 53 tree-ring samples, which were considered for ring measurements (Table 4). Tree-ring measurements revealed younger ages, with a maximum of 136 years in a pear tree. Walnut and service trees reached a maximum of 105 and 86 years, respectively, while apple trees were characterized by maximum tree-ring measured ages below 50 years (Table 4).

**Table 4.** Statistics of fruit trees analyzed through dendrochronology methods. The cores without stem pith (asterisks) were not used for dendrochronology analysis.

| Species | No. Trees (Total) | No. Trees (No Pith) | DBH (cm) | | | Age (Years) | | | Elevation (m a.s.l.) | | |
|---|---|---|---|---|---|---|---|---|---|---|---|
| | | | Mean | Max | Min | Mean | Max | Min | Mean | Max | Min |
| Chestnut | 9 | 5 * | 74 | 102 | 56 | 82 | 94 | 49 | 404 | 618 | 210 |
| Cherry | 12 | 3 * | 63 | 80 | 42 | 50 | 96 | 22 | 803 | 945 | 586 |
| Mulberry | 7 | - | 62 | 107 | 38 | 57 | 100 | 30 | 565 | 853 | 329 |
| Apple | 7 | 4 * | 31 | 35 | 24 | 44 | 42 | 46 | 639 | 820 | 454 |
| Walnut | 3 | 1 * | 68 | 80 | 56 | 90 | 105 | 75 | 563 | 819 | 308 |
| Pear | 27 | 4 * | 51 | 94 | 20 | 80 | 136 | 41 | 707 | 1089 | 25 |
| Service | 4 | - | 49 | 57 | 43 | 81 | 86 | 73 | 482 | 776 | 52 |

*3.2. Growth and Longevity of Fruit Trees*

Trees with DBH below 50 cm had both the youngest and the oldest ages measured through tree ring analyses (Figure 4a). Mean radiocarbon ages of trees with DBH below 100 cm spanned from 160 to 646 years (Figure 4b). Radiocarbon ages were more dispersed than tree ring ages (Figure 4b). Tree age and stem DBH showed no correlation when tree-ring and radiocarbon ages were analyzed separately (Figure 4a,b), but a significant positive correlation was observed by merging the two datasets (Figure 4c). When data were analyzed by species, only pear trees showed a significant positive correlation between age and DBH (Figure 5a). Of these pear trees, 12 were dated through radiocarbon analyses, half of which had ages after 1950, and the other half had ages from 160 to 467 years (Table 2, Figure 6b). Pear tree ages were positively correlated with elevation (Figure 5b), but annual mean growth was not (Figure 5c).

Olive trees were dated only through radiocarbon analyses due to the presence of false rings and intra-annual wood density fluctuations which prevented reliable tree ring measurements [27]. A total of 3 out of 14 olive trees were radiocarbon dated after 1950 (Table 2), while the remaining 11 trees showed mean ages spanning from $212 \pm 130$ to $680 \pm 57$ years, with no relationship between DBH and age (Figure 6a). Radiocarbon analyses showed that chestnut (1 tree) and mulberry (4 trees) had ages ranging from about two to seven centuries (Table 2, Figure 6c).

Ring width measures were used to reconstruct growth histories (annual DBH; Figure 7). Stem increment trajectories highlighted the higher growth rates of younger trees compared with older trees, whose ages were determined through radiocarbon analyses (Figure 7).

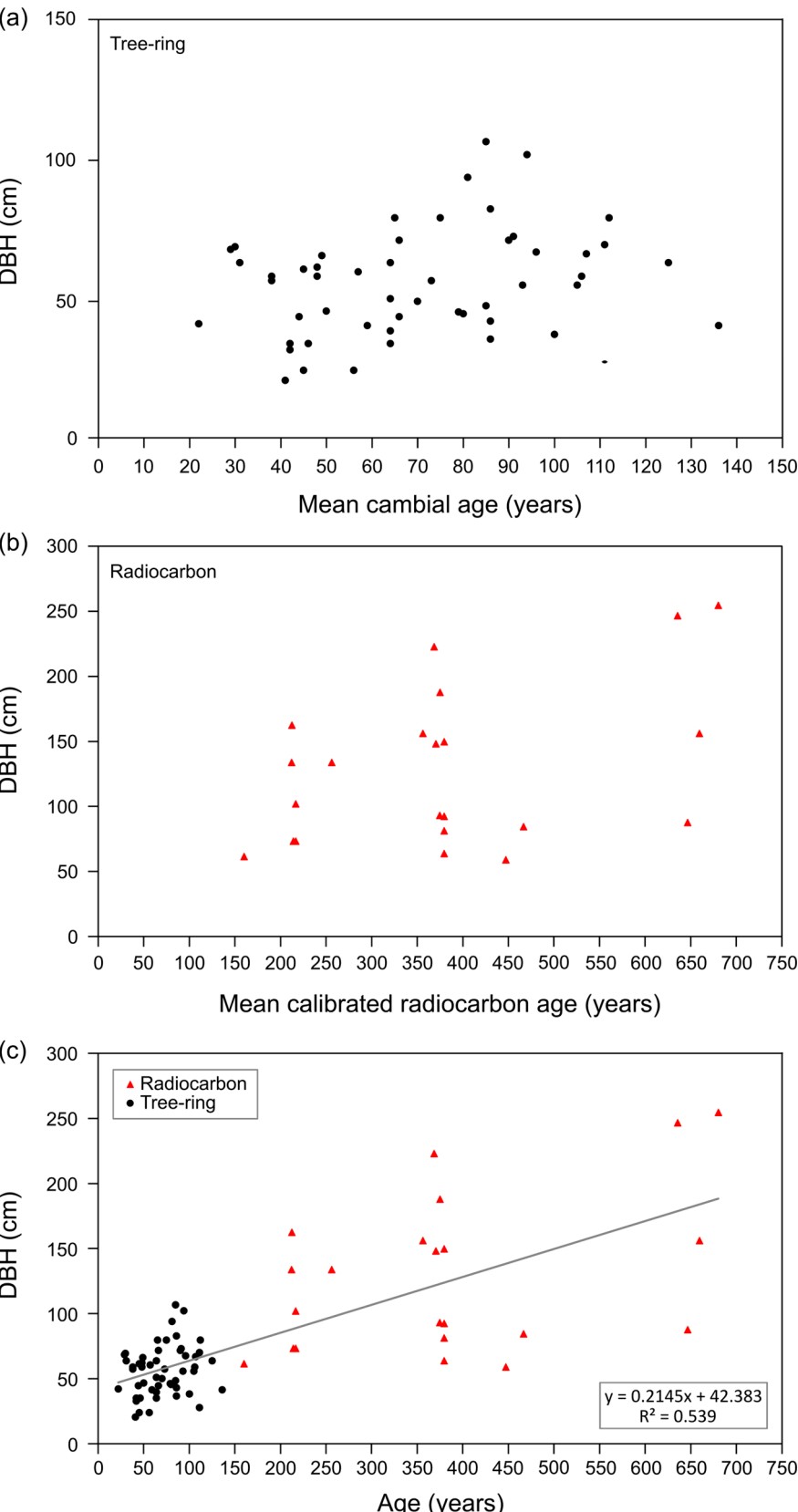

**Figure 4.** Age vs. diameter (DBH) relationships of fruit trees in Basilicata. (**a**) Age measured with tree-ring methods. (**b**) Age estimated with radiocarbon. (**c**) Total sample size (tree-ring and radiocarbon estimates); the linear regression line, equation, and $R^2$ value are shown.

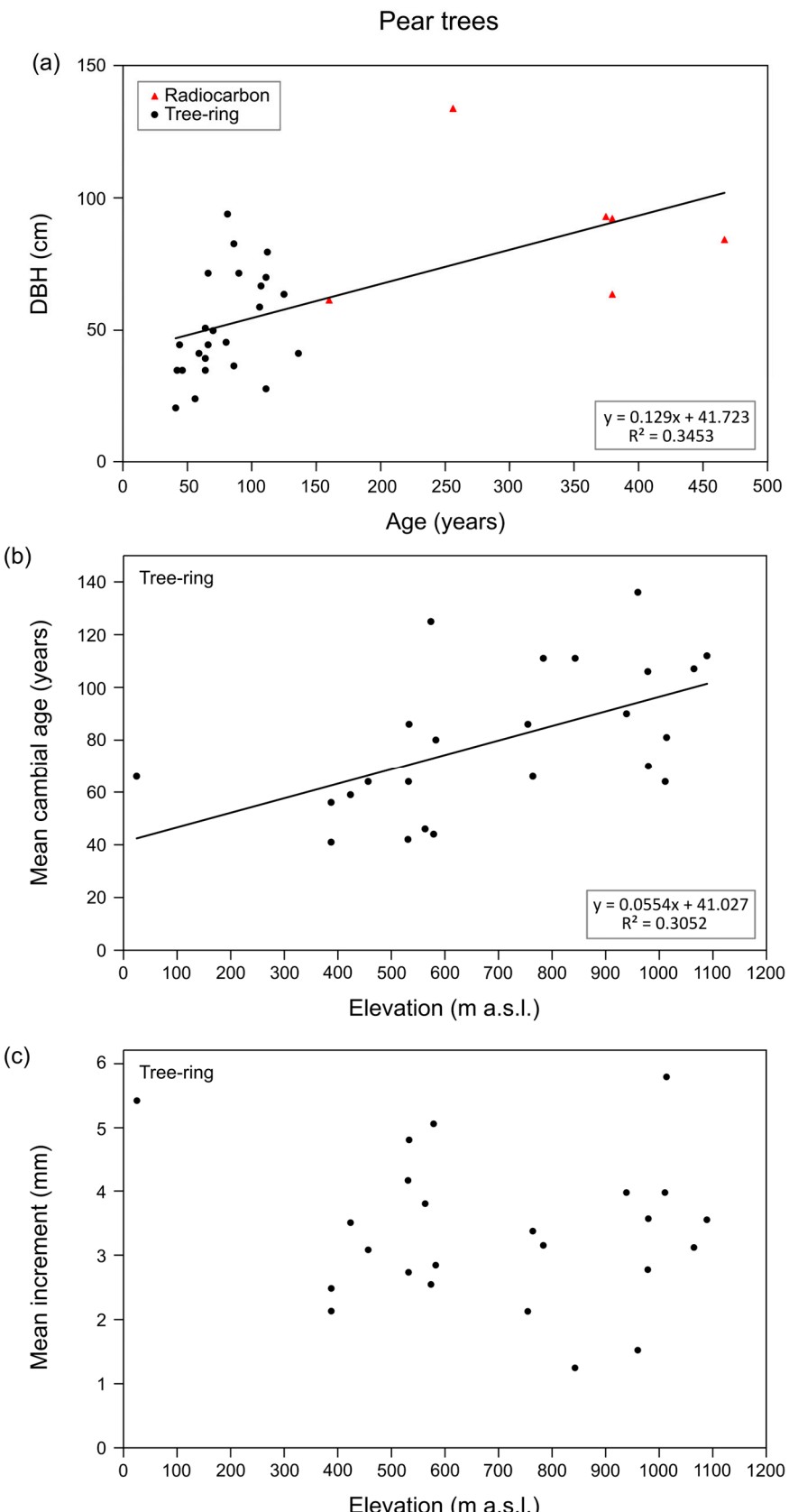

**Figure 5.** Focus on pear trees; linear regression and $R^2$ value are displayed when the linear correlation was statistically significant. (**a**) Age vs. diameter (DBH). (**b**) Age (tree-ring only) vs. elevation. (**c**) Mean radial increment vs. elevation.

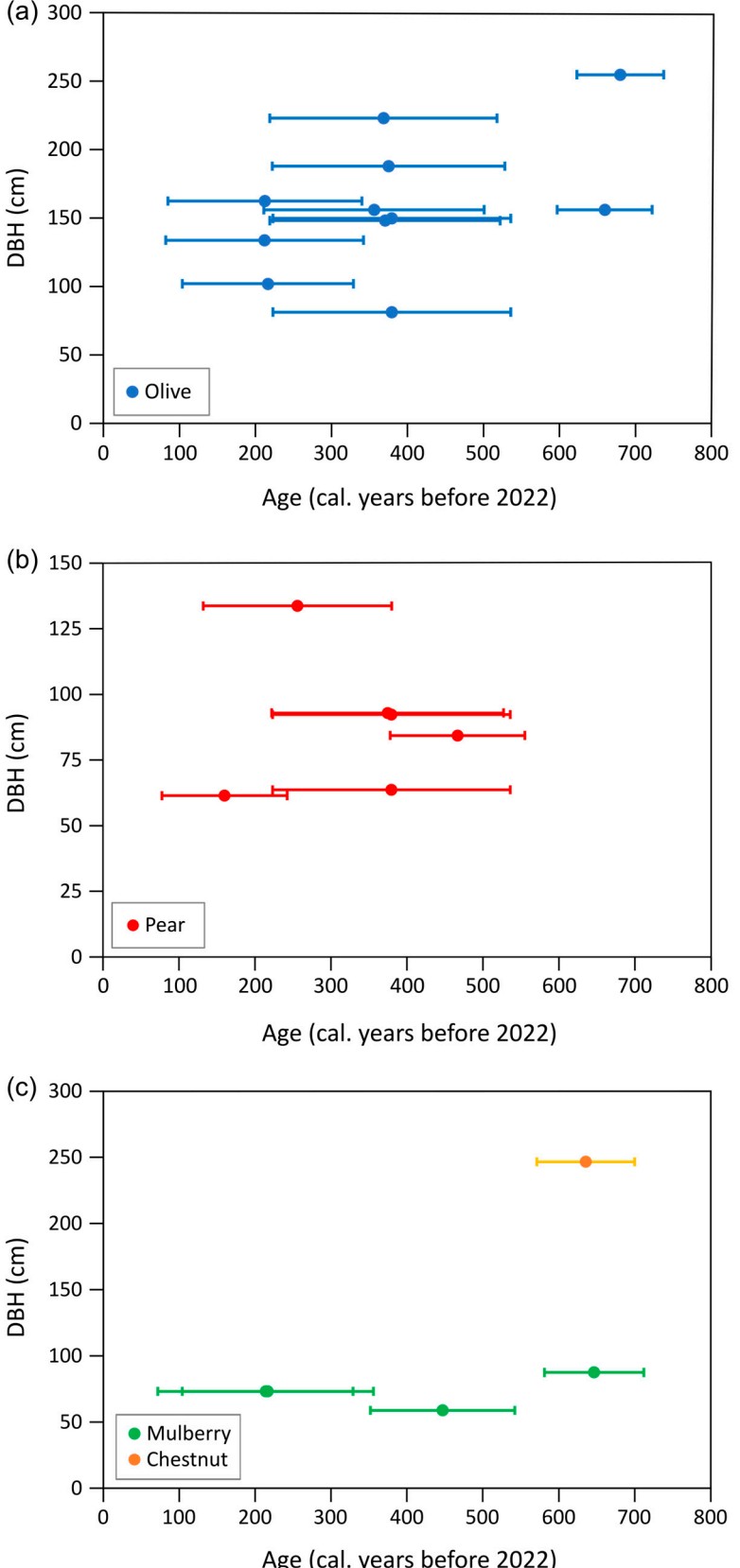

**Figure 6.** Radiocarbon analysis results for (**a**) Olive trees, (**b**) Pear trees and (**c**) Mulberry and Chestnut trees. Raw radiocarbon dates were calibrated (see Section 2) and ages were calculated from the sampling year (2022). Confidence intervals (2 sigma) of calibrated radiocarbon dates are shown as horizontal bars.

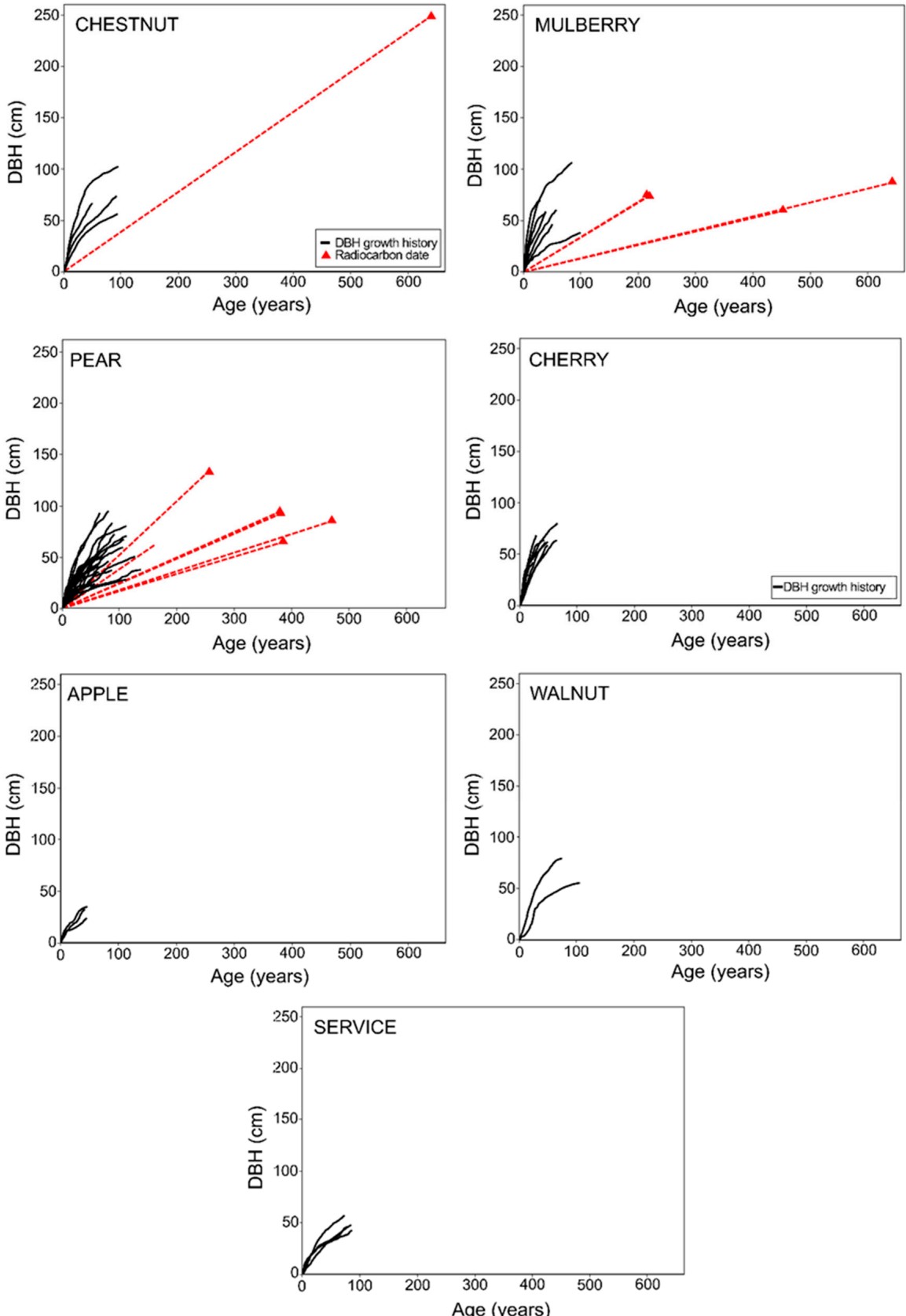

**Figure 7.** DBH growth histories (black lines) calculated from annual ring-width measured through dendrochronological methods. Radiocarbon ages of chestnut, mulberry and pear trees (red triangles) were included for reference and used to simulate linear growth (dashed red lines).

## 4. Discussion

### 4.1. The Age of Fruit Patriarchs in Basilicata

We demonstrated that ancient fruit trees can be successfully dated using tree ring and radiocarbon dating methods, see also [21] for an integrated approach to the study of ancient wild trees. Tree ring analyses are still underutilized in arboriculture despite recent studies [15,16,28,29]. Some fruit species are nearly impossible to examine through tree ring analysis, e.g., olive trees [27], so radiocarbon analyses are needed to estimate the age of monumental individuals [30,31].

The fruit tree heritage of Basilicata showed remarkable ages compared to the global mean age of broadleaves, which has been estimated at 229 years from public domain dendrochronological data [32], and at 334 years when also including radiocarbon dated trees [33]. In Basilicata, olive, chestnut and mulberry trees showed a potential longevity comparable to wild broadleaves, which in the Italian Apennines can reach up to one millennium [34].

The oldest fruit tree in our dataset was an olive of Ferrandina, which turned out to be $680 \pm 57$ years old (Figure 4a). Olive trees of similar age were found in the Garden of Gethsemane (Jerusalem, Israel) [30] and in Malta [35]. Other studies involving monumental olive trees in Spain and Israel have not found such remarkable ages [28,31]. Our data seems to confirm that the age of olive trees is often over-estimated based on morphological features, which do not consider complex growth histories and trajectories [28].

Monumental chestnut trees are often considered pluri-millennial individuals in the popular press, but there was no scientific support for these claims in our study. The chestnut of Lagonegro (Figure 3c), whose age was estimated at $636 \pm 66$ years, is still remarkable for the species. On the basis of growth models, [9] estimated an age of 700 years for two chestnuts with large diameter (318 cm) in the Swiss Alps. In the British islands, a veteran chestnut of 300–350 years old had a diameter ranging from 111 to 286 cm [29]. The contributing role of slow growth rates for maximizing longevity [34] was confirmed by the age/size relationship of the chestnut of Lagonegro, which has a relatively less impressive diameter of 247 cm.

Mulberry is regarded as a long-lived tree species in Asia, where anecdotal reports describe sacred individuals exceeding one thousand years of age [36]. In Britain, the oldest mulberry trees were dendrochronologically dated to approximately 150 years old [16]. Mulberry trees of Basilicata show that this species can become remarkably old in southern Europe. The oldest mulberry tree of our dataset was aged to $647 \pm 66$ years, suggesting that in Italy, the species was introduced long before the late Medieval times, likely during the Byzantine period [7].

Pear species originated in the southwest regions of China and are among the oldest fruit crops in the world [37,38]. The pear germplasm in the Italian varieties can be traced back to the 1st century of the Roman Empire, as documented by Pliny the Elder [39,40]. Their wide commercial appreciation, nutritional importance, and adaptability to various environments [41] allowed pear trees to be spread by humans outside of their native area. Pear trees were introduced to North America during early colonization, and since then, they have become a widespread fruit tree, especially in the northwest (see https://usapears.org/history-of-pears/, accessed on 20 January 2023). In Danvers, Massachusetts, the Endicott Pear Tree, planted around 1630, is considered the oldest living cultivated tree in North America [42]. Our census found a high number of old pear trees growing in Basilicata whose ages can exceed 450 years (Table 2), especially in mountainous areas.

### 4.2. Factors That Favor Old Ages in Fruit Trees

The main factor behind tree longevity is most likely a taxonomic one. Olive, chestnut, mulberry, and pear trees are characterized by extractive-rich duramen (heartwood), and the capacity for canopy rejuvenation following severe disturbances such as pruning and storms. Similarly to what happens to wild trees, these traits are inevitably linked to extreme longevity [34]. Their long lifespan and ability to survive under adverse conditions (biotic

and abiotic risk factors) suggests that these trees have adapted over many years to their original habitat. In our study, we confirmed the limited longevity of apple and cherry trees, whose maximum lifespan recorded to date barely exceeds a century. Other potentially long-lived fruit trees for which we had limited or no data are service and walnut trees, which may, however, reach extended lifespans [43].

Each old and ancient tree represents a unique genetic resource that brings useful information about resistance to natural biotic and abiotic conditions, biomass and fruit productivity, and synthesis of phytochemical molecules with nutraceutical potential [44]. Moreover, long lifespans may have contributed to enriching the genome of somatic and reproductive tissue cells not only with mutations of DNA bases [45], but also by inducing the accumulation of epigenetic mutations. Such additive memory could then turn each veteran tree into a potentially rich node of unique relationships with other organisms and environmental conditions [17,33,46,47].

Stem and crown sizes are not predictive of tree age in wild trees [32,34]. A linear relationship between stem DBH and age emerged only by combining tree ring dated trees and old radiocarbon-measured ages (Figure 4c). Dendro-dated trees were mature individuals, while radiocarbon dated tree were either mature (born before 1950 CE) or ancient (Table 3). Combining both mature and ancient trees resulted in a linear relationship between size and age, whereas the data points were highly dispersed when the two categories were treated separately, especially in the case of radiocarbon dated trees (Figure 4a,b). This implies that equations for estimating tree age based on stem size are likely unreliable, because local environmental conditions and management practices determine different growth histories. The absence of a linear relationship between age and DBH in our data (Figures 4a,b and 6) suggests that in fruit trees, as it happens in forests, dimensions are not predictive of tree age.

Local environmental conditions, including light intensity, photoperiod, UV intensity, air and soil temperature, humidity and precipitation, soil fertility and stability, affect the growth and survival of fruit trees in rural cultural landscapes, influencing age-size relationships along natural gradients, e.g., elevation, within the same species (Table 1, Figure 5b). Therefore, these factors not only impact morphometric features but also physiology and metabolism in ways that shape their life cycles, and ultimately define the size and lifespan of a tree [48,49].

In the case of fruit trees, in addition to environmental factors, the cultivation system (e.g., pruning, tillage, watering, fertilization, harvesting) plays a role in their long-term growth, thereby generating heterogeneous responses (Figure 5c). In this regard, it is interesting to note how the centuries-old mulberry trees were characterized by a much lower growth rate than the younger trees (Figure 7), probably due to frequent pruning/collection of branches or other disturbances in the past. In pear trees, however, current and past growth trajectories were not too different, as shown by a lower departure of the tree ring derived growth trajectories compared to the linearly simulated ones based on radiocarbon dates (Figure 7).

Monumental fruit trees are rarely located in intensive or extensive agricultural spaces and monocultures (Table 2). Ancient individuals are more commonly an element characterizing historical agricultural landscapes with high environmental heterogeneity. Ancient olives, for example, were not mapped in large orchard characterized by productive varieties and mechanized management schemes. In the case of chestnuts, monumental trees are often found in seminatural environments where they have survived a series of diseases and pests (e.g., ink sickness, cortical cancer) which have often led to the transformation of fruit chestnut groves into coppices.

### 4.3. The Importance of Fruit Tree Longevity for Conservation Biology and Genetic Resources

Century-old trees could be considered an active hub of biological diversity [17,50]. Ancient fruit trees provide ecological niches for several plant and animal species of high conservation value [51–53]. Despite their limited distribution over agrarian landscapes,

these "habitat trees" enhance local biodiversity in ways that would otherwise be unavailable [54]. Their preservation is, therefore, crucial for ecosystems capable of providing vital services for agriculture, such as pollinating insects. Monumental trees of great age add value to human-dominated landscapes, both as living witnesses of past historical changes and for their esthetic and recreational contributions [55]. Maintaining a healthy inventory of such trees over the landscape is a mark of success for any program aimed at sustainable rural development.

Climate change is considered a relevant threat that fruit tree crops in Mediterranean environments will face in the near future. Modern cultivars are poorly adapted to shorter winters, late-spring frost, increased drought spells and hot waves, sudden changes in temperature, abnormal rainfall, salinization of soils and other environmental anomalies. Ancient fruit trees contain, in their own genomic and epigenomic structure, usable information on the adaptive strategies that enabled them to survive adverse conditions [45]. As climatic changes simultaneously interacted with other agents, from microbes to soils, trees evolved signaling systems to detect surrounding environmental factors and alien organisms. The acquired information is biologically integrated, thus generating plastic and adaptive physiological responses for plant survival [56,57]. Therefore, the capacity to implement and/or modify strategies against enemy organisms [47], which has allowed century-old trees to survive and expand their ramifications above and below ground, becomes necessarily intertwined with the way in which space is occupied by the developing crown and root architecture, simultaneously monitoring and leading the growth of its organs in space in relation to the environment and to its ontogenetic development.

Most of the old trees analyzed are in good vegetative condition and still bear fruits. Their preservation is ensured by national regulation aimed at protecting monumental trees in the Italian landscape (Law No. 10/2013 and DM 23 October 2014). Three of the individuals included in this study (one olive and two chestnut) are listed in the "register of monumental trees of Basilicata". The other centuries-old individuals, including pears and mulberries of monumental size, will be proposed for the inclusion in the national register. Moreover, most of the studied monumental trees belongs to genetic resources listed in the "register of agrobiodiversity resources of Basilicata" (Regional Law No. 26, 14/10/2008 and D.M. No. 39407, 9 December 2019). Furthermore, the genetic maintenance of these old-fruit trees is today promoted by the Lucan Agency for the Development and Innovation in Agriculture (ALSIA) through agamic propagation. Once propagated, these ancient lines of fruit plants will re-enter historical landscapes to maximize the chance that old trees will be present in the future. Maintaining an old-fruit tree stock requires active management not only to protect them but also to propagate them. At the same time, a didactic program entitled "adopt a patriarch" was launched to inform the public about the value of ancient fruit trees for a sustainable future. This study represents a key step towards conserving their unique cultural and natural heritage in the context of sustainable management and restoration of agricultural landscapes.

### 4.4. Future Research

Additional studies on centuries-old fruit trees will be interdisciplinary, given their relevance for both natural ecological processes and human-dominated systems. The application of comparative genomics to ancient ecotypes, for example, could provide opportunities for investigating mutations favorable to tree survival under abiotic and biotic stressors (e.g., drought, pathogens) and for agronomic traits selected during past centuries (e.g., fruit size, load, quality, etc.). Comparative genomic analysis of the tissues of diverse organs in centuries-old plants may also reconstruct and reveal which traits were selected during adverse events that occurred in their lifetime, such as the Little Ice Age. Furthermore, comparative genomics analyses can be correlated with dendrochronology and radiocarbon analyses to understand when mutations occurred and to unravel their significance in terms of adaptation [58]. In case of grafted ancient trees or, more generally, vegetative propagation, the availability of clonal lines separated from others by centuries allows the

study of mutation, micro-evolutive and aging processes of meristematic cells (e.g., in the cambium). Unravelling this knowledge means understanding levels of interaction not expressed by the phenotypes of cultivated plants. A highly significant scenario would be that of a science-based design of resilient fruit tree portfolios in the Mediterranean region, with the long-term goal of conserving old tree heritage for genetic improvement and for selecting varieties that will work best in future orchards.

This study was a first step toward a comprehensive understanding of the distribution and longevity of fruit trees in the Mediterranean. Future research including individuals growing in other cultural landscapes and under different climate conditions is needed to disentangle the relative contribution of species-specific traits, environmental factors and human actions to expressed longevity and survival of ancient fruit trees.

## 5. Conclusions

In the Basilicata region, we uncovered some of the oldest scientifically dated fruit trees in the world. These veterans, which characterize the cultural landscape from the coast to the mountain belt since the late Medieval time, may hold a unique genetic resource. They crossed the Little Ice Age and are surviving current climate change as they fend off pests and diseases. Furthermore, their large architecture and time persistence guarantee ecological niches and micro-habitats suitable for flora and fauna species.

In the agrarian landscapes of the Basilicata region, old and ancient fruit trees are not only a reminder of the past, since they are still actively producing resources. Our investigation has provided information needed for effective management policies that can guarantee their persistence and regeneration, and their transfer to upcoming generations. Social initiatives such as the "adopt a patriarch" program have the twin function of conserving the unique genetic heritage of old-fruit trees and promoting on-the-ground biodiversity conservation strategies in rural areas. Conservation measures aimed at preserving cultural landscapes and their traditional elements are crucial to oppose industrial agriculture and to sensitize local communities to the ecological management of agricultural spaces. Old-fruit tree conservation can be used as a sustainable indicator of a more harmonic relationship between nature and humans.

**Author Contributions:** Conceptualization, G.P., R.M., D.C. and P.Z.; methodology, G.P., L.C., G.Q. and D.C.; software, M.B., G.Q. and F.S.; validation, M.B., M.D., J.P. and G.Q.; formal analysis, M.B., F.S., J.P., L.C., G.Q. and M.D.; data curation, M.B., J.P., F.S., D.C., N.S., A.S., P.Z., G.Q., M.D. and G.P.; writing—original draft preparation, J.P., G.P., R.M. and F.B.; writing—review and editing, J.P., M.B., F.B., L.C., D.C., M.D., R.M., A.S., G.Q., N.S., F.S., P.Z. and G.P.; visualization, J.P., M.B. and F.S.; supervision, G.P., R.M. and F.B.; funding acquisition, G.P. and R.M. All authors have read and agreed to the published version of the manuscript.

**Funding:** The research project on ancient fruit trees was financed, in part, by the project FiNoPom "Caratterizzazione morfogenetica e conservazione delle varietà autoctone di fico, nocciolo e pomacee della Basilicata" sottomisura 10.2 PSR Basilicata 2014–2020. J.P. was supported, in part, by MIUR (Ministry for Education, University and Research) initiative Department of Excellence (Law 232/2016). F.B. was supported, in part, by the Experiment Station of the College of Agriculture, Biotechnology and Natural Resources at the University of Nevada, Reno, NV, USA.

**Data Availability Statement:** The data presented in this study are available on request from the corresponding author.

**Acknowledgments:** We are grateful to the landowners, farmers, and citizens, altogether too many to list by name, who have helped us with locating potentially old fruit trees.

**Conflicts of Interest:** The authors declare no conflict of interest.

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
