# Peer review of "The Longevity of Fruit Trees in Basilicata (Southern Italy): Implications for Agricultural Biodiversity Conservation"

_land, doi:10.3390/land12030550_

Round 1

Reviewer 1 Report

Dear Authors,

The topic under discussion is important, but in its current form, it will not arouse any broader interest. In my opinion your manuscript does not suits the profile of the journal and the Special Issue. It is very important that you uncovered some of the oldest scientifically dated fruit trees in the world.  But it is only one step, which should be described in other journal, eg. Horticulturae.

Unfortunately in its current form, the article does not bring valuable conclusions to scientific knowledge, what is confirmed by the general conclusions. Here are some tips that could improve the quality of the article.

I missed a well-formulated aim, which would facilitate the description of the results and drawing conclusions. The formulation of a hypothesis would also be welcome.

Also the novelty of the work against the background of previous articles should be clearly emphasized.

Management recommendations should be more precise. It should be explained in detail on what basis they were formulated

I suggest to use a diagram for describing the methods (e.g. a flow chart).

Figure 3. – please add legend describing points.

Which lessons learned could be used in other European regions. What are the limitation of your research ?

Best regards and stay healthy!

Author Response

Dear Referee 1,

Thanks for your observations and suggestions that were useful to improve our work.

In this reviewed version of the manuscript, we added a clear list of aims, which helps clarify the novelty of the work, the choice of the methodology applied and how our methods worked to fill in current gaps in the age determination of fruit trees around the world. Please find below the new text (ll 79-93):

“Here, we present the distribution and attributes of 106 living old-fruit trees in southern Italy (Basilicata), some of which have survived over multiple centuries. Age of monumental fruit trees is often mystified and/or assumed on non-scientific bases, such as historical narratives, portraits, or other anecdotal data. Determining the age of monumental fruit trees with scientific methods is challenging because of rotten wood, growth anomalies and false rings. In this study, we aim at (i) determining stem ages of mature and old fruit trees with integrated tree-ring and radiocarbon methods that overcome difficulties related to monumental sizes and hollowed tree stems; (ii) assessing growth patterns of old fruit trees with respect to younger ones; (iii) describing the environment in which old fruit trees grow; and (iv) discussing implications for conservation biology and sustainable management of agricultural lands. We also argue why and how to conserve their genotypes through specific initiatives that involve local communities, and we present the ongoing efforts promoted by the Lucan Agency for the Development and Innovation in Agriculture (ALSIA) to disseminate the value of ancient varieties of fruit trees for a sustainable future”.

We edited Figure 3 by adding the missing legend.

Also, we edited the conclusion section and made more references to the nature of the management recommendations proposed, and their meaning in contrast to the global ecological crisis. Also, we better explained strategies that can apply to other rural contexts, both European and non-European. Please find below the edited text in the conclusion section (ll 494-503):

“Our investigation has provided information needed for effective management policies that can guarantee their persistence and regeneration, and their transfer to incoming generations. Social initiatives such as the “adopt a patriarch” program have the twin function of conserving the unique genetic heritage of old-fruit trees and promoting on-the-ground biodiversity conservation strategies in rural areas. Conservation measures aimed at preserving cultural landscapes and their traditional elements are crucial to oppose industrial agriculture and to sensitize local communities to the ecological management of agricultural spaces. Old fruit tree conservation can thus be used as a sustainable indicator of a more harmonic relationship between nature and humans”.

Reviewer 2 Report

The research is well structures and the design is very solid, supporting the results and the conclusions.

This is a specialised area of research but for those in the sector it is very worthwhile and important research and would also have some appeal to those who are not directly involved in the sector or discipline. On this basis it is very much worth publication.

The paper is well structured and very easy to follow and read. The results are logical and well explained.

Author Response

Dear Referee 2,

We are very thankful for your positive assessment.

Reviewer 3 Report

Dear colleagues,

I thank you for having the opportunity to read this manuscript that brings into discussion a topical issue, namely the longevity of fruit trees and the implications for the conservation of agricultural biodiversity in southern Italy.

I would have only one recommendation: to introduce a literature review chapter, in which you can identify how this subject was approached in other Mediterranean regions of the world. Thus, to identify what is missing from the specialized literature and to mention more clearly what your study brings. In other words, what can you say about the research gap?

Sincerely,

Author Response

Dear Referee 3,

Thanks for your positive assessment and the suggestion of making more references to other studies on fruit trees longevity. In this new version of the manuscript, we edited the last part of the introduction section by adding more references to knowledge gaps and difficulties in determining the age of fruit trees, and a clear list of aims (ll 80-89):

“Age of monumental fruit trees is often mystified and/or assumed on non-scientific bases, such as historical narratives, portraits, or other anecdotal data. Determining the age of monumental fruit trees with scientific methods is challenging because of rotten wood, growth anomalies and false rings. In this study, we aim at (i) determining stem ages of mature and old fruit trees with integrated tree-ring and radiocarbon methods that overcome difficulties related to monumental sizes and hollowed tree stems; (ii) assessing growth patterns of old fruit trees with respect to younger ones; (iii) describing the environment in which old fruit trees grow; and (iv) discussing implications for conservation biology and sustainable management of agricultural lands.”.

 Although there is not a dedicated literature review chapter, we performed a literature review on the longevity of fruit trees around the world and referenced these studies in the discussion (sub. par. 4.1 “The age of fruit patriarchs in Basilicata”).

In this new version of the manuscript, we added two more statements referring to what is still missing in the specialized literature: ll 427-430 “Climate change is considered a relevant threat that fruit tree crops in Mediterranean environments will face in the near future. Modern cultivars are poorly adapted to shorter winters, late-spring frost, increased drought spells and hot waves, sudden changes in temperature, abnormal rainfall, salinization of soils and other environmental anomalies”.; ll 482-485 “A highly significant scenario would be that of a science-based design of resilient fruit tree portfolios in the Mediterranean region, with the long-term goal of conserving the old tree heritage for genetic improvement and for selecting varieties that will work best in future orchards”.

We also edited the conclusion section and made more references to the contribution of our study to the topic of the Special Issue (ll 493-503):

“In the agrarian landscapes of the Basilicata region, old and ancient fruit trees are not only a reminder of the past, since they are still actively producing resources. Our investigation has provided information needed for effective management policies that can guarantee their persistence and regeneration, and their transfer to incoming generations. Social initiatives such as the “adopt a patriarch” program have the twin function of conserving the unique genetic heritage of old-fruit trees and promoting on-the-ground biodiversity conservation strategies in rural areas. Conservation measures aimed at preserving cultural landscapes and their traditional elements are crucial to oppose industrial agriculture and to sensitize local communities to the ecological management of agricultural spaces. Old fruit tree conservation can thus be used as a sustainable indicator of a more harmonic relationship between nature and humans”.

Reviewer 4 Report

The research entitled "Longevity of fruit trees in Basilicata (Southern Italy): Implications for the conservation of agricultural biodiversity" is discussed.

Very interesting article with applicable results and some impact. The reinterpretation of the history of fruit trees and their historical use based on the results obtained is very interesting. It opens up new avenues of research in relation to genomic value.
The following suggestions are made:
MATERIALS AND METHODS
The materials and methods section should be revised and expanded to better understand the research and its scientific basis:
- It is suggested to include a description of the geographical context in which the research is framed. Physical-environmental characteristics (topography, altitude, lithology, general climate and vegetation) and also the structure of the property, type of land use, landscape, etc.
- Make a list or table with the parameters that have been taken into account when obtaining the information from the surveys carried out, including units of measurement for each parameter and thresholds for delimitation and classification if applicable.
- Line 2.1. It is suggested to better explain the procedure for the elaboration of surveys: how many were carried out, in what context, for how long, at what time of the year.
RESULTS
Lines 124 to 128. Suggested to move to methodology.
Figure 1. It is suggested to include the scientific name of the identified trees in the legend.
What is the significance of the data collected with respect to: environmental context (e.g., urban, peri-urban, agricultural); slope; topographic exposure; soil rockiness and soil depth and profile? Explain what these data have been collected for and their results. They can be included in a matrix with all the information of the sampled specimens. Some questions on these issues can also be answered. A characterisation of the specimens based on these collected data could be very interesting, for example: Where are these specimens located (urban, peri-urban, etc.)? Are there differences in the physical-environmental conditions? What is their state of conservation? Etc.
DISCUSSION
Line 228-229: Revise the statement that there are only two species of Pyrus in Europe. In principle more than 10 are recognised.

Other questions that may be of interest to raise in the paper:

- Are the trees privately or publicly managed?
- Are they endowed with any kind of recognition or protection. If so, it would be interesting to include this information in the above-mentioned characterisation section.
- Are there historical documents on the use to which these or similar trees have been put in their geographical context? What kind of use do they have today? In which agricultural context are they situated? Are they part of boundaries or at the centre of properties?
- Do they have any identity value among the local population?

Author Response

Dear Referee 4,

We are grateful for your specific and valuable comments and suggestions that helped improve the completeness and clarity of this research paper. 

We made the following edits in this revised version of the manuscript:

The previous Lines 124 to 128 have been partially moved to the Materials and Methods section;

The previous lines 228-229 have been removed;

We added to the Result section two Tables (the new Table 1 & 2) which displays the topographical and land cover setting of the trees under investigation. In the Tables, a summary of elevation (mean, max, min), slope % (mean, max, min), Aspect % (in the 8 cardinal directions) and the land-use/land-cover category derived from the Corine Land Cover. This data is divided per tree species. The new data is presented in the Result section (ll 153-188):

“Fruit trees were distributed from the lowland to the mountain elevation belt, and from plain topography to steep slopes. Most of them were concentrated in the warmest aspects – i.e., from SE to SW (Table 1). Pear trees were found over the largest elevation range, from 43 to 1097 m a.s.l., and on the steepest slopes (63%; Table 1). Also, pears were the only trees located in all aspects, including the coldest ones, i.e. N, NE and NW (Table 1). Other fruit species were absent on north-exposed land (Table 1). Cherry, pear and walnut were found above 700 m a.s.l. on average (Table 1). Olives were recorded at lower mean elevation – 362 m a.s.l. – although some individuals were found up to 800 m a.s.l. (Table 1).

Most old fruit trees were located in heterogeneous agricultural areas, which include complex cultivation patterns, forming a mosaic landscape where cultivated lands mix with semi-natural ecosystems (Table 2). Olive and mulberry trees could also be found in the discontinuous urban fabric category (Table 2). Old olive and chestnut trees were rarely located in olive groves and chestnut stands, and many chestnut trees lie in the shrub and/or herbaceous vegetation association (Table 2). Cherry, walnut and service trees were often recorded in forests or transitional woodland categories (Table 2)”. 

We also discussed the significance of monumental olives and chestnut individuals in respect of their land use context in the Discussion (ll 399-406):

“Monumental fruit trees are rarely located in intensive or extensive agricultural spaces and monocultures (Table 2). Ancient individuals are more commonly an element characterizing historical agricultural landscapes with high environmental heterogeneity. Ancient olives, for example, were not mapped in large orchards characterized by productive varieties and mechanized management schemes. In the case of chestnuts, monumental trees are often found in seminatural environments where they have survived a series of diseases and pests (e.g., ink sickness, cortical cancer) which have often led to the transformation of fruit chestnut groves into coppices”.

We could not include in the methods section a specifical description of the geographical context in which the research is framed, because the study was carried out in a large and environmentally heterogeneous region of southern Italy, which cannot be described through general topography, altitude, lithology, climate and vegetation features. We choose to extrapolate this information on single tree points and summarize the data per tree species in Tables 1 & 2.

We added more specifics to the current conservation measures applied to monumental trees in Italy, which included some of the trees object of this study, and the future conservation measures that will be applied to the ancient fruit patriarchs of Basilicata. Please find below the added text (ll 442-450):

“Most of the old trees analyzed are in good vegetative condition and still bear fruits. Their preservation is ensured by national regulation aimed at protecting monumental trees in the Italian landscape (Law No. 10/2013 and DM 23 October 2014). Three of the individuals included in this study (one olive and two chestnuts) are listed in the “register of monumental trees of Basilicata”. The other centuries-old individuals, including pears and mulberries of monumental size, will be proposed for inclusion in the national register. Moreover, most of the studied monumental trees belong to genetic resources listed in the “register of agrobiodiversity resources of Basilicata” (Regional Law No .26, 14/10/2008 and D.M. No. 39407, 9 December 2019)”.

Round 2

Reviewer 1 Report

Dear Authors,

Thank you for taking into account some of my suggestions and for the effort you put into improving the text. Personally, I still expect the following changes to be made in the text:

1.     Please use a diagram for describing the methods (e.g. a flow chart).

2.     Please describe the limitation of your research, taking into account environmental conditions (resulting from, among others, location).

Best regards!

Author Response

Dear Referee,

Thanks for your positive assessment of the edits we made to the first version of the manuscript.

In this new revised version, we decided to follow both of your recommendations. We added a flow chart describing the methods (new Figure 2 and attached to the present Reply) and insert a statement on the limitation of the present study and what we should address in future research. Please find below the new statement (ll 568-572):

"This study was a first step toward a comprehensive understanding of the distribution and longevity of fruit trees in the Mediterranean. Future research including individuals growing in other cultural landscapes and under different climate conditions is needed to disentangle the relative contribution of species-specific traits, environmental factors and human actions to expressed longevity and survival of ancient fruit trees".

Kind regards
